# Robust Textual Embedding against Word-level Adversarial Attacks

**Yichen Yang***      **Xiaosen Wang***      **Kun He**†

School of Computer Science and Technology, Huazhong University of Science and Technology, Wuhan, China
`{yangyc,xiaosen,brooklet60}@hust.edu.cn`

## Abstract

We attribute the vulnerability of natural language processing models to the fact that similar inputs are converted to dissimilar representations in the embedding space, leading to inconsistent outputs, and we propose a novel robust training method, termed *Fast Triplet Metric Learning (FTML)*. Specifically, we argue that the original sample should have similar representation with its adversarial counterparts and distinguish its representation from other samples for better robustness. To this end, we adopt the triplet metric learning into the standard training to pull words closer to their positive samples (*i.e.*, synonyms) and push away their negative samples (*i.e.*, non-synonyms) in the embedding space. Extensive experiments demonstrate that FTML can significantly promote the model robustness against various advanced adversarial attacks while keeping competitive classification accuracy on original samples. Besides, our method is efficient as it only needs to adjust the embedding and introduces very little overhead on the standard training. Our work shows great potential of improving the textual robustness through robust word embedding.

## 1 INTRODUCTION

Deep learning models have achieved impressive performance on various machine learning tasks [Krizhevsky et al., 2012, Devlin et al., 2019], however, recent studies have shown their vulnerability to adversarial examples crafted by exerting elaborate and imperceptible perturbations on the original input data. Adversarial examples are firstly found in models for image classification task [Szegedy et al., 2014, Goodfellow et al., 2015], and recently tremendous attention

has been attracted in various natural language processing (NLP) tasks [Papernot et al., 2016, Wang et al., 2022].

For adversarial attacks on text classification, the character-level perturbations [Gao et al., 2018, Ebrahimi et al., 2018] could be easily eliminated by a spell checker [Pruthi et al., 2019], while the sentence-level attacks [Iyyer et al., 2018, Wang et al., 2020] based on rephrasing are hard to preserve the original semantics. In contrast, the word-level attacks [Ren et al., 2019, Alzantot et al., 2018, Zang et al., 2020, Wang et al., 2021c, Maheshwary et al., 2021] based on synonym substitutions have become the most widely adopted approach as they could craft the adversarial examples with high success rate while maintaining the grammatical correctness and semantic consistency, which are more challenging to defend against.

To improve the model robustness against the word-level adversarial attacks, researchers have implemented adversarial training by crafting adversarial examples with their proposed attacks and incorporating the adversaries into the training set [Ren et al., 2019, Alzantot et al., 2018]. However, due to the discrete input space of the texts, the adversarial attacks for texts take much longer time compared with those for images and could not generate sufficient adversarial examples at each training epoch. To this end, Wang et al. [2021c] and Dong et al. [2021] make improvement by designing fast white-box adversarial attacks to speed up adversarial training, yet it still takes dozens of times longer than the standard training. Certified defense methods based on interval bound propagation (IBP) [Jia et al., 2019, Huang et al., 2019] could provide theoretical lower bound on the robustness. However, due to the heavy computing overhead and strict constraints, it is hard to extend the certified methods to large-scale datasets and complex models such as BERT. Meanwhile, a recent progress [Wang et al., 2021b] maps synonyms to the same code to eliminate perturbation in the input space, which is efficient and easy-to-apply.

In this work, we regard the texts obtained by synonym substitutions on the original sample as *similar samples*. We

---

*Equal Contribution.

†Corresponding author.

*Accepted for the 38th Conference on Uncertainty in Artificial Intelligence* (UAI 2022).

attribute the vulnerability of deep learning models to the dissimilar representations of similar input samples, among which those ultimately leading to wrong predictions are adversarial examples. For robustness, we propose a *Fast Triplet Metric Learning (FTML)* to force each word in the input text to be close to its synonyms and far away from the non-synonyms in the embedding space. In this way, any original sample would have similar representations with its adversarial counterparts crafted by synonym substitution based attacks and distinguish its representation from other samples in the embedding space. By incorporating the triplet metric learning into the standard training, we could train a robust model against the synonym substitutions based adversarial attacks.

Our main contributions are summarized as follows:

- FTML is the first adversarial defense approach that focuses on robust word embedding. It reveals a new perspective of enhancing the NLP model robustness, highlighting the difference of adversarial defense between the text domain and image domain.

- In FTML, we propose a general idea to learn a robust word embedding by pulling words closer to their synonyms while pushing non-synonyms further away in the embedding space. Such general idea of how to learn a robust word embedding is important to defend against word-level adversarial attacks.

- Extensive experiments demonstrate that FTML could significantly promote the model robustness against various advanced adversarial attacks while keeping high accuracy on original data across multiple datasets and models, including CNN, LSTM and BERT.

- FTML is also efficient, because it introduces only a little overhead to the standard training for adjusting the embedding, facilitating its application to large-scale datasets and complex models.

## 2 RELATED WORK

### 2.1 ADVERSARIAL ATTACKS

According to different types of perturbation, existing adversarial attacks fall into three categories: (a) Character-level attacks usually utilize character insertion, modification, or deletion to craft adversarial perturbation [Gao et al., 2018, Ebrahimi et al., 2018, Li et al., 2019]. (b) Sentence-level attacks are based on rephrasing [Iyyer et al., 2018, Zhang et al., 2019b] or inserting short related sentences [Wang et al., 2020, Liang et al., 2018]. (c) Word-level attacks substitute words in the input text with their synonyms according to different strategies [Papernot et al., 2016, Zhang et al., 2019a, Wang et al., 2021c, Meng and Wattenhofer, 2020, Jin et al., 2020]. Probability Weighted Word Saliency

(PWWS) [Ren et al., 2019] greedily determines the substitutions based on both the output probability change and the word saliency. Genetic Attack (GA) [Alzantot et al., 2018] and Particle Swarm Optimization (PSO) [Zang et al., 2020] employ population-based optimization algorithms to search for adversarial examples. By only querying the predicted label, Hard Label Attack (HLA) [Maheshwary et al., 2021] crafts adversarial examples through random substitutions and reduces the perturbation using a genetic algorithm.

### 2.2 ADVERSARIAL DEFENSES

Three categories of textual defense methods have been proposed to boost the model robustness against word-level attacks: adversarial training based methods, certified defense methods, and input transformation based methods.

Adversarial training (AT) is one of the most popular defense methods [Goodfellow et al., 2015, Madry et al., 2018, Alzantot et al., 2018, Ren et al., 2019, Ivgi and Berant, 2021]. Adversarial Training with FGPM enhanced by Logit pairing (ATFL) [Wang et al., 2021c] utilizes their proposed FGPM to generate adversarial examples and injects them into the training set. Adversarial Sparse Convex Combination (ASCC) [Dong et al., 2021] utilizes a weighted combination of the word vectors of synonyms to replace the original word vector, and then optimizes the weights by a gradient optimization to craft virtual adversarial examples in the embedding space for adversarial training.

Certified defense methods provide the models with provable robustness to all possible word substitutions [Zeng et al., 2021, Wang et al., 2021a, Huang et al., 2019]. Given the interval of input, Jia et al. [2019] utilize Interval Bound Propagation (IBP) to calculate the upper and lower bound of the output layer by layer, and then minimize the worst-case loss that any combination of the word substitutions can induce to achieve certified robustness.

Input transformation based methods eliminate the adversarial perturbations in the input space. Wang et al. [2021b] propose Synonym Encoding Method (SEM) that inserts a coder before the input layer to map all the synonyms to the same code. SEM is efficient and easy-to-apply without involving the model architecture and training process.

Different from SEM [Wang et al., 2021b] that assigns the same code to synonyms in the input space, our work delicately captures the synonymous and non-synonymous relations of words by adjusting the distances among the words in the embedding space. We will provide detailed analysis on the differences between SEM and our work in Section 4.5. Besides, since we do not need to craft adversarial examples during the training process and only introduce a little calculation, compared with typical adversarial training or certified defense methods, our method is easier to be extended to large-scale datasets and complex models.

# 3 METHODOLOGY

This section first formulates the adversarial examples for text classification, then introduces our motivation, and finally describes the proposed defense method.

## 3.1 PRELIMINARY

Let $\mathcal{X}$ denote the input space containing all the texts, and $\mathcal{W}$ the dictionary containing all legal words in the input texts. Let $\mathcal{Y} = \{y_1, y_2, \cdots, y_c\}$ denote the output space containing all the classification labels. Given an input text with $n$ words $x = \langle w_1, w_2, \cdots, w_n \rangle$ where $w_i \in \mathcal{W}$, a classifier $f : \mathcal{X} \to \mathcal{Y}$ firstly encodes $x$ into the sequence of word vectors denoted as $v(x) = \langle v(w_1), v(w_2), \cdots, v(w_n) \rangle$ in the embedding layer, and then feeds the representations $v(x)$ into subsequent layers to output the prediction label $f(x)$, which is expected to be the true label $y$.

Next, we define the synonyms and the adversarial examples based on synonym substitutions. Following the previous works [Jia et al., 2019, Wang et al., 2021b, Dong et al., 2021], we define the synonym set $\mathcal{S}(w)$ as no more than $k$ nearest words of $w$ within the Euclidean distance $\delta$ in the third-party GloVe embedding space post-processed by counter-fitting technique [Mrkšić et al., 2016].

To achieve the semantic consistency, the attacker adopts synonym substitutions to craft the adversarial example $x' = \langle w_1', w_2', \cdots, w_n' \rangle, w_i' \in \mathcal{S}(w_i) \cup \{w_i\}$, such that:

$$f(x') \neq f(x) = y, \quad s.t. \ R(x, x') \leq \epsilon, \quad (1)$$

where $R(x, x')$ is the distance metric of two texts in the input space, and $\epsilon$ is a small constant used to constrain the distance between $x$ and $x'$. For the word level adversarial examples, we adopt the word substitution ratio as the distance metric $R(x, x_{adv})$ in the input space:

$$R(x, x') = \frac{1}{n} \sum_{i=1}^{n} \mathbb{1}_{w_i \neq w_i'}(w_i, w_i'), \quad (2)$$

where $\mathbb{1}$ is the indicator function.

## 3.2 MOTIVATION

Although the adversarial example $x'$ is close to the original sample $x$ in the input space, it succeeds to deceive the classifier. We attribute the vulnerability of the classifier to the fact that $x$ and $x'$ have rather remote representations, leading to inconsistent outputs. In contrast, a robust classifier should be able to extract similar representations when feeding with similar input samples.

To this end, we adopt the triplet metric learning, which is commonly used for machine learning algorithms where an input as the anchor sample is compared to its positive

samples and negative samples. A straightforward way is to regard the original text $x$ as the anchor sample, the similar texts $x'$ obtained by synonym substitutions as the positive sample, and other text $\tilde{x}$ sampled from the dataset as the negative sample. Given a triplet $\langle x, x', \tilde{x} \rangle$, the triplet loss is formulated as:

$$\mathcal{L}(x, x', \tilde{x}) = \max \{d(x, x') - d(x, \tilde{x}) + \alpha, 0\}, \quad (3)$$

where $d(\cdot, \cdot)$ denotes the distance metric between the representations of two samples, and $\alpha$ is a margin between the distance of positive and negative pairs. Minimizing $\mathcal{L}(x, x', \tilde{x})$ forces the model to extract representations for the original sample $x$, which is similar to that of its similar samples $x'$ but dissimilar from that of other samples $\tilde{x}$.

However, exploring the positive samples is a combinatorial optimization problem, and the time complexity grows exponentially with the text length. To solve this issue, we turn to a word-level solution. The combinations of words are hard to exhaust, but the synonyms of words are limited. In the embedding space, if words are forced to be close to their synonyms and far away from non-synonyms, then any input text will have similar representations with its potential adversarial examples crafted by synonym substitutions and distinguish their representations from that of other samples in the dataset. Thus, we propose Fast Triplet Metric Learning (FTML) which uses triplet loss to adjusting word embedding for robustness.

## 3.3 FAST TRIPLET METRIC LEARNING

We first formulate the word-level triplet loss, and then describe how to incorporate the triplet loss with standard training to train a robust model.

**Word-level Triplet Loss**. For two words $w_a$ and $w_b$, we use the $\ell_p$-norm distance of their word vectors in the embedding space as the distance metric between them:

$$d(w_a, w_b) = \|v(w_a) - v(w_b)\|_p. \quad (4)$$

In this work, we adopt the Euclidean distance, *i.e.*, $p = 2$. Then, we design the triplet loss for a word $w$ as follows:

$$\mathcal{L}_{tr}(w, \mathcal{S}(w), \mathcal{N}) = \frac{1}{|\mathcal{S}(w)|} \sum_{w' \in \mathcal{S}(w)} d(w, w') - \frac{1}{|\mathcal{N}|} \sum_{\tilde{w} \in \mathcal{N}} \min(d(w, \tilde{w}), \alpha) + \alpha, \quad (5)$$

where $\mathcal{S}(w)$ denotes the synonym set of word $w$, and $\mathcal{N}$ the set containing words randomly sampled from the dictionary. The number of randomly sampled words is the same as the maximum number of synonyms, namely $k$. We minimize $\mathcal{L}_{tr}(w, \mathcal{S}(w), \mathcal{N})$ to decrease the distances between the word $w$ and its synonyms (positive samples) and increase the distances between $w$ and its non-synonyms (negative

samples) in the embedding space. In addition, to prevent the distance of positive pairs and negative pairs from keeping increasing simultaneously, the negative pairs would no longer be pushed away once the distance exceeds $\alpha$.

**Overall Training Objective**. Given a text $x = \langle w_1, w_2, \cdots, w_n \rangle$ with the ground-truth class label $y$ as the current input, we formulate the overall training objective as follows:

$$\mathcal{L}(x,y) = \mathcal{L}_{ce}(f(x), y) + \beta \cdot \frac{1}{n} \sum_{i=1}^{n} \mathcal{L}_{tr}(w_i, \mathcal{S}(w_i), \mathcal{N}_i), \quad (6)$$

where $\mathcal{L}_{ce}(\cdot, \cdot)$ denotes the cross-entropy loss, and $\beta$ is a hyper-parameter to control the weight of the triplet loss. The first term $\mathcal{L}_{ce}$ is used to train the subsequent layers after the first embedding layer for the classification's capability. The second term $\mathcal{L}_{tr}$ is designed to train a robust word embedding, where each word in the input text is forced to be close to its synonyms and far away from the non-synonyms in the embedding space. In this way, we could train a robust model that has similar representations for similar input samples and distinguishes representations from that of other samples in the dataset, alleviating the vulnerability exploited by attackers while maintaining good classification performance.

Note that the metric learning is performed only in the first embedding layer of NLP models, regardless of the subsequent architecture of models. For BERT models that incorporate various input representations, including word embeddings, segment embeddings, and position embeddings, we only perform our metric learning on word embeddings. Furthermore, FTML is fast because adjusting the embedding introduces only a little overhead to the standard training. Hence, theoretically FTML is generic to any NLP models.

## 4 EXPERIMENTS

This section evaluates FTML with four defense baselines against various attacks on three benchmark datasets involving CNN, LSTM and BERT models. Code is available at `https://github.com/JHL-HUST/FTML`.

### 4.1 EXPERIMENTAL SETUP

**Datasets.** We evaluate FTML on three benchmark datasets, namely *IMDB* [Maas et al., 2011], *Yelp-5* and *Yahoo! Answers* [Zhang et al., 2015]. *IMDB* is a binary sentiment classification dataset containing 25,000 movie reviews for training and 25,000 for testing. *Yelp-5* has 640,000 training samples and 50,000 testing samples with five labels. *Yahoo! Answers* is a large-scale topic classification dataset with 10 classes, consisting of 1,400,000 training samples and 50,000 testing samples.

**Models**. We replicate CNN [Kim, 2014] and bidirectional LSTM [Liu et al., 2016] from Wang et al. [2021c]. For

brevity, we denote the bidirectional LSTM with LSTM. We use the 300-dimensional GloVe word vectors [Pennington et al., 2014] to initialize the embedding layer of CNN and LSTM models. We also fine-tune the base-uncased pre-trained BERT [Devlin et al., 2019] as another target model.

**Attack Methods.** To thoroughly evaluate the defense efficacy of the FTML defense methods, we adopt four advanced adversarial attacks, including GA [Alzantot et al., 2018], PWWS [Ren et al., 2019], PSO [Zang et al., 2020], and HLA [Maheshwary et al., 2021]. Due to the inefficiency of textual adversarial attacks, we attack each model using 1000 randomly sampled testing data from each dataset to craft adversarial examples.

**Defense Baselines**. We compare FTML with standard training and four state-of-the-art adversarial defense baselines, including a certified defense: IBP [Jia et al., 2019], an input transformation based defense: SEM [Wang et al., 2021b], and two adversarial training based defenses: ATFL [Wang et al., 2021c] and ASCC [Dong et al., 2021]. However, as indicated in Shi et al. [2020], BERT models are too challenging to be tightly verified with current IBP technologies, neither do Wang et al. [2021c] extend their adversarial training method ATFL to BERT. Thus, we omit IBP and ATFL as the defense baselines on BERT models.

**Evaluation Settings**. For the synonym definition in Section 3.1, we follow Jia et al. [2019] and Dong et al. [2021] and set $k = 8$ and $\delta = 0.5$ for all the experiments to have a fair comparison. For the hyper-parameters in FTML, we set the margin $\alpha = 0.7\alpha_0$ in Eq. 5, where $\alpha_0$ is the average word distance of the initial word embeddings before training. $\alpha_0$ is $8.54$ for CNN and LSTM models initialized by GloVe word vectors and $1.48$ for the base-uncased pre-trained BERT model. We set the weight $\beta = 1$ in Eq. 6 to achieve a proper trade-off between the standard training loss and the triplet metric loss. We will provide a hyper-parameter study to explore their sensitivity. We train our models for 20 epochs on *IMDB*, but $5$ epochs on *Yelp-5* and *Yahoo! Answers* respectively, as the models converge faster on large datasets.

### 4.2 EVALUATION ON DEFENSE EFFICACY

**Performance on CNN and LSTM.** We compare FTML with standard training (Standard) and four defense baselines, IBP, ATFL, SEM and ASCC, using the original samples (Clean) or the adversarial examples crafted by different attack methods. The comparison results are presented in Table 1. The more effective the defense method, the higher the classification accuracy under various attacks. Meanwhile, we also wish the performance do not decay much on the original samples, as compared with the standard training.

From the results, we can observe that FTML achieves dominant robustness across all datasets under various adversarial

Table 1: The classification accuracy (%) against various adversarial attacks on three datasets for CNN and LSTM. The columns of *Clean* denote the classification accuracy on the entire original testing set. The highest accuracy against the corresponding attack on each column is highlighted in **bold**, while the second one is highlighted in underline. The last row of each block indicates the gains of the accuracy between FTML and the best baseline.

| Dataset | Defense | CNN | | | | | LSTM | | | | |
|---|---|---|---|---|---|---|---|---|---|---|---|
| | | Clean | PWWS | GA | PSO | HLA | Clean | PWWS | GA | PSO | HLA |
| *IMDB* | Standard | 89.7 | 0.6 | 2.6 | 1.4 | 17.7 | 89.1 | 0.2 | 1.6 | 0.3 | 8.7 |
| | IBP | 81.7 | 75.9 | 76.0 | 75.9 | 76.6 | 77.6 | 67.5 | 67.8 | 67.6 | 68.2 |
| | ATFL | 85.0 | 63.6 | 66.8 | 64.7 | 72.8 | 85.1 | 72.2 | 75.5 | 74.0 | 77.7 |
| | SEM | 87.6 | 62.2 | 63.5 | 61.5 | 70.5 | 86.8 | 61.9 | 63.7 | 62.2 | 70.8 |
| | ASCC | 84.8 | 74.0 | 75.5 | 74.5 | 77.6 | 84.3 | 74.2 | 76.8 | 75.5 | 79.5 |
| | FTML | 88.1 | **81.1** | **81.4** | **81.1** | **82.4** | 87.2 | **79.0** | **79.2** | **78.8** | **79.7** |
| | | | ↑5.2 | ↑5.4 | ↑5.2 | ↑4.8 | | ↑4.8 | ↑2.4 | ↑3.3 | ↑0.2 |
| *Yelp-5* | Standard | 62.7 | 1.1 | 1.3 | 0.8 | 1.1 | 64.8 | 0.5 | 0.9 | 0.4 | 0.5 |
| | IBP | 52.1 | 47.8 | 47.8 | 47.7 | 47.6 | 42.6 | 42.1 | 40.7 | 40.6 | 40.2 |
| | ATFL | 61.4 | 50.0 | 51.7 | 50.2 | 53.9 | 62.4 | 48.0 | 48.8 | 46.9 | 51.9 |
| | SEM | 60.1 | 34.9 | 33.8 | 32.4 | 37.2 | 61.9 | 35.5 | 34.3 | 33.7 | 37.6 |
| | ASCC | 58.9 | 47.3 | 49.3 | 47.4 | 50.6 | 59.9 | 48.5 | 50.5 | 49.6 | 52.5 |
| | FTML | 59.9 | **56.7** | **56.7** | **56.6** | **56.5** | 61.9 | **57.5** | **57.6** | **57.5** | **57.6** |
| | | | ↑6.7 | ↑5.0 | ↑6.4 | ↑2.6 | | ↑9.0 | ↑7.1 | ↑7.9 | ↑5.1 |
| *Yahoo! Answers* | Standard | 72.6 | 6.8 | 7.2 | 4.9 | 7.0 | 74.7 | 12.2 | 9.6 | 6.5 | 10.4 |
| | IBP | 63.1 | 54.9 | 54.9 | 54.8 | 55.0 | 54.3 | 47.3 | 47.6 | 47.0 | 47.3 |
| | ATFL | 72.5 | 62.5 | 63.1 | 62.5 | **65.0** | 73.6 | 61.7 | 60.8 | 60.3 | 63.1 |
| | SEM | 70.1 | 53.8 | 52.4 | 51.9 | 54.6 | 72.3 | 57.0 | 56.1 | 55.4 | 56.8 |
| | ASCC | 69.0 | 58.4 | 59.6 | 58.5 | 59.9 | 70.7 | 61.7 | 62.3 | 61.9 | 63.2 |
| | FTML | 69.4 | **65.1** | **65.1** | **65.0** | 64.9 | 71.4 | **67.8** | **67.8** | **67.8** | **67.9** |
| | | | ↑2.6 | ↑2.0 | ↑2.5 | ↓0.1 | | ↑6.1 | ↑5.5 | ↑5.9 | ↑4.7 |

attacks with clear margins. Since HLA attack is based on the hard-label setting and is easier to defend against, the superiority of FTML is relatively insignificant when compared to other defense methods under HLA attack. In addition, among the defense methods, FTML yields the best clean accuracy on *IMDB* dataset and is close to the best clean accuracy on the other two datasets.

**Performance on BERT.** Before using FTML to fine-tune the BERT model, we need to make some modifications to the vocabulary of the pre-trained BERT model. Some words in the training set do not exist in the vocabulary, and will be separated into multiple tokens by the WordPiece tokenizer adopted by the BERT model, making it hard to calculate the distance involving these words. Thus, we add these words to the vocabulary of BERT, and initialize their embedding vectors by the average pre-trained embedding vectors of the tokens that they would have been split into, so that each word corresponds to a single embedding vector. We will investigate the effect of supplementing BERT's vocabulary on vocabulary size, robustness, and generalization of BERT models in Section 4.5.

The results on three datasets are shown in Table 2. Similar to the performance on CNN and LSTM, FTML exhibits the

best robustness over the state-of-the-art defense baselines by a large margin. Besides, for the clean accuracy, FTML decays the least on *IMDB*, and is only inferior to SEM by less than $1\%$ on the other two datasets. Note that we evaluate the defenses on a more aggressive attack setting that the nearest $k = 8$ words satisfying the distance constraint $\delta$ are regarded as synonyms, and therefore the robustness results of reproduction are lower than that reported in the SEM paper [Wang et al., 2021b] where $k = 4$.

### 4.3 EVALUATION ON DEFENSE EFFICIENCY

The efficiency is also crucial for evaluating the defense methods, especially when a defense is applied on large-scale datasets and complex models, such as BERT model on *Yahoo! Answers* dataset. The training time cost per epoch for the models with various defense methods is shown in Table 3. The IBP defense heavily slows the training on LSTM models due to the high overhead for certified constraints. Although designed with fast white-box adversarial attacks, the two adversarial training based methods, ATFL and ASCC, are still not easy to scale to large datasets and complex models. SEM is the most efficient approach be-

Table 2: The classification accuracy (%) against various adversarial attacks on three datasets for BERT models. As explained in subsection 4.1, we omit IBP and ATFL as the baselines on BERT models.

| Dataset | Defense | Clean | PWWS | GA | PSO | HLA |
|---|---|---|---|---|---|---|
| *IMDB* | Standard | 92.4 | 16.6 | 8.1 | 1.9 | 8.2 |
| | SEM | 89.9 | 72.3 | 70.5 | 69.2 | 75.2 |
| | ASCC | 81.3 | 65.1 | 65.4 | 63.1 | 69.5 |
| | FTML | 91.3 | **81.2** | **81.5** | **80.0** | **83.1** |
| | | | ↑8.9 | ↑11.0 | ↑10.8 | ↑7.9 |
| *Yelp-5* | Standard | 65.7 | 2.4 | 1.1 | 0.7 | 1.3 |
| | SEM | 63.7 | 39.9 | 37.0 | 36.9 | 39.4 |
| | ASCC | 63.4 | 50.0 | 50.8 | 49.5 | 54.8 |
| | FTML | 63.0 | **55.4** | **55.1** | **55.0** | **55.2** |
| | | | ↑5.4 | ↑4.3 | ↑5.5 | ↑0.4 |
| *Yahoo! Answers* | Standard | 77.0 | 20.7 | 10.3 | 7.3 | 10.0 |
| | SEM | 75.6 | 64.6 | 62.0 | 61.9 | 63.6 |
| | ASCC | 75.2 | 66.4 | 67.5 | 66.6 | 68.0 |
| | FTML | 74.8 | **70.0** | **70.0** | **70.0** | **70.0** |
| | | | ↑3.6 | ↑2.5 | ↑3.4 | ↑2.0 |

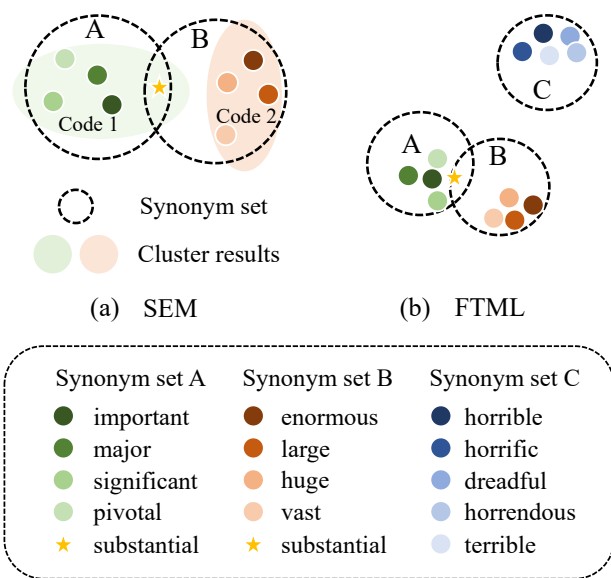

(a)  SEM        (b)  FTML

Synonym set — dashed circle
Cluster results — shaded circle

Synonym set A: important, major, significant, pivotal, ★ substantial
Synonym set B: enormous, large, huge, vast, ★ substantial
Synonym set C: horrible, horrific, dreadful, horrendous, terrible

Figure 1: Illustration of the differences between SEM and FTML. Since word *substantial* is polysemous, both synonym set $A$ and $B$ contain *substantial*, and they are semantically related but not really synonymous. Synonym set $C$ is not semantically related to the previous two synonym sets.

cause it only needs to transform the input text based on a synonym coding. Since we introduce a little overhead for adjusting the word embeddings on the standard training, our proposed FTML needs slightly longer time for the training than SEM, but ours is at least four times faster than other defense methods on LSTM and BERT models. Especially on the BERT model, the extra cost of adjusting the embedding is less than 10% of the time spent on standard training, which is almost negligible. It is worth consuming a little longer time for gaining a much better defense efficacy.

## 4.4   EVALUATION ON WORD EMBEDDING

To validate that FTML learns a robust word embedding, we apply the standard training on the models using frozen word embeddings initialized by three types of pre-trained word vectors respectively: (a) GloVe word vectors [Pennington et al., 2014]. (b) Word vectors of base-uncased pre-trained BERT without fine-tuning [Devlin et al., 2019], denoted as BERT-V. (c) Robust word vectors trained by FTML on three models using the corresponding dataset, denoted as CNN-RV, LSTM-RV, and BERT-RV, respectively. Note that during the training process, we freeze the initialized word vectors and only do standard training to check whether our trained word embeddings are beneficial to the model robustness.

As shown in Table 4, the standard trained models initialized with our robust word vectors could efficaciously block the attacks without any other defense mechanisms, and even perform better than all the defense baselines (see results in Table 1 and Table 2), manifesting the great potential of improving the robustness of NLP models through word em-

beddings. Furthermore, our robust word embeddings exhibit good defense transferability across the models. For instance, CNN-RV pre-trained on CNN model yields high robustness on LSTM model, and vise versa for LSTM-RV. The transferability indicates that once we have trained a robust word embedding, we can easily apply it to other models, and only require standard training for the classification's capability.

## 4.5   FURTHER ANALYSIS

Here we provide further analysis on the differences of FTML and SEM, as well as an FTML variant based on contrastive learning. We also discuss on the influence of different $\ell_p$-norm distance metrics and hyper-parameters. In addition, we investigate the effect of supplementing BERT's vocabulary with actual words on robustness and generalization.

**Analysis on FTML and SEM.** Among the baselines, SEM is the most similar approach to ours. However, SEM directly maps the synonyms to the same code in the input space, while FTML adjusts the representations by metric learning in the embedding space. For SEM, there is no discrimination for words assigned the same code, and no relation between words with different codes. On the contrary, FTML could use the distance metric in the embedding space to delicately capture the semantic relationship among the words.

For the instance in Figure 1 (a) for SEM, all words inside the green cluster is mapped to one code, while all words inside the red cluster is mapped to another code, and there is no

Table 3: Evaluation on the training time per epoch (in minutes) for the models with various defenses.

| Defense | IMDB | | | Yelp-5 | | | Yahoo! Answers | | |
|---|---|---|---|---|---|---|---|---|---|
| | CNN | LSTM | BERT | CNN | LSTM | BERT | CNN | LSTM | BERT |
| Standard | 1 | 1 | 11 | 4 | 6 | 178 | 9 | 13 | 371 |
| IBP | 1 | 48 | N/A | 12 | 610 | N/A | 26 | 953 | N/A |
| ATFL | 15 | 22 | N/A | 203 | 290 | N/A | 444 | 573 | N/A |
| SEM | 1 | 1 | 11 | 4 | 6 | 178 | 9 | 16 | 371 |
| ASCC | 2 | 6 | 100 | 32 | 71 | 1638 | 55 | 125 | 2523 |
| FTML | 1 | 1 | 12 | 12 | 16 | 183 | 22 | 29 | 384 |

Table 4: The classification accuracy (%) of models initialized and frozen with different word vectors without any other defenses on *IMDB* dataset.

| Model | Embedding | Clean | PWWS | GA | PSO | HLA |
|---|---|---|---|---|---|---|
| CNN | GloVe | 89.3 | 0.9 | 2.8 | 1.5 | 18.3 |
| | CNN-RV | 87.3 | 79.9 | 80.7 | 80.0 | 81.1 |
| | LSTM-RV | 87.0 | 79.2 | 79.6 | 79.2 | 80.2 |
| LSTM | GloVe | 89.4 | 1.9 | 4.0 | 1.4 | 17.5 |
| | CNN-RV | 87.9 | 80.2 | 80.8 | 80.3 | 81.9 |
| | LSTM-RV | 87.8 | 80.7 | 81.0 | 80.9 | 81.8 |
| BERT | BERT-V | 92.5 | 22.0 | 12.6 | 8.2 | 7.0 |
| | BERT-RV | 90.3 | 79.7 | 80.4 | 79.0 | 82.5 |

Table 5: The classification accuracy (%) against various adversarial attacks of models trained with contrastive metric learning (CML) on *IMDB* dataset.

| Model | Method | Clean | PWWS | GA | PSO | HLA |
|---|---|---|---|---|---|---|
| CNN | CML | 88.3 | 78.6 | 79.1 | 78.8 | 81.1 |
| | FTML | 88.1 | 81.1 | 81.4 | 81.1 | 82.4 |
| LSTM | CML | 87.2 | 76.5 | 76.5 | 76.0 | 78.2 |
| | FTML | 87.2 | 79.0 | 79.2 | 78.8 | 79.7 |
| BERT | CML | 92.1 | 71.9 | 67.3 | 64.1 | 63.3 |
| | FTML | 91.3 | 81.2 | 81.5 | 80.0 | 83.1 |

Table 6: The classification accuracy (%) of models trained with FTML involving different $L_p$-norm distance metric on *IMDB* dataset.

| Model | Distance | Clean | PWWS | GA | PSO | HLA |
|---|---|---|---|---|---|---|
| CNN | $p = 1$ | 88.5 | 76.9 | 77.9 | 77.6 | 80.1 |
| | $p = 2$ | 88.1 | 81.1 | 81.4 | 81.1 | 82.4 |
| | $p = \infty$ | 88.8 | 29.0 | 39.4 | 30.4 | 58.1 |
| LSTM | $p = 1$ | 88.0 | 75.8 | 76.5 | 75.8 | 77.7 |
| | $p = 2$ | 87.2 | 79.0 | 79.2 | 78.8 | 79.7 |
| | $p = \infty$ | 88.3 | 35.2 | 34.1 | 25.1 | 49.0 |
| BERT | $p = 1$ | 91.6 | 80.3 | 80.4 | 79.0 | 83.4 |
| | $p = 2$ | 91.3 | 81.2 | 81.5 | 80.0 | 83.1 |
| | $p = \infty$ | 92.6 | 57.5 | 45.2 | 40.9 | 22.2 |

making the representations between similar samples closer. To verify whether contrastive learning can also enhance the robustness by pulling words closer to their synonyms and pushing away their non-synonyms in the embedding space, we replace the triplet loss $\mathcal{L}_{tr}$ with the contrastive loss $\mathcal{L}_{ct}$, which could be formulated as follows:

$$\mathcal{L}_{ct}(w, \mathcal{S}(w), \mathcal{N}) = -\log \frac{\sum_{w' \in \mathcal{S}(w)} \exp(-d(w, w')/\tau)}{\sum_{\tilde{w} \in \mathcal{S}(w) \cup \mathcal{N}} \exp(-\min(d(w, \tilde{w}), \alpha)/\tau)}, \quad (7)$$

where $\tau$ is the temperature hyper-parameter. We test the $\tau$ in $\{1, 5, 10, 15, 20, 25, \cdots, 55, 60\}$ on CNN model with other settings being unchanged, and choose $\tau = 20$ for the best robustness. We denote this variant as Contrastive Metric Learning (CML). As shown in Table 5, CML is inferior to FTML on the three models consistently, especially on the BERT model. Note that CML also significantly boosts the model's robustness compared to the standard trained models, and has competitive performance with the defense baselines (see results in Table 1 and Table 2), indicating that our motivation of adjusting word distances in the embedding space is sound.

**Analysis on Distance Metric**. We adopt the Euclidean distance of word vectors by default to define the word distance, that is, $p = 2$ in Eq. 4. To explore the effect of different $\ell_p$-norm distance metrics on FTML, we also evaluate the defense efficacy of FTML with Manhattan distance ($p = 1$)

relations between the two codes. SEM distributes *substantial* to the green cluster and loses synonymous relationship between *substantial* and other words in synonym set $B$.

As illustrated Figure 1 (b) for FTML, although after the training, word *substantial* is more closer to other words in the synonym set $A$ with an average distance of $0.0004$ in the embedding space. Compared to the average distance of $6.69$ between *substantial* and the semantically unrelated synonym set $C$, FTML has also forced *substantial* to be closer to other words in the other synonym set $B$ with the average distance of $3.85$. Benefiting from the above property, FTML performs better than SEM empirically.

**Variant based on Contrastive Metric Learning.** Contrastive learning is popular recently with the purpose of

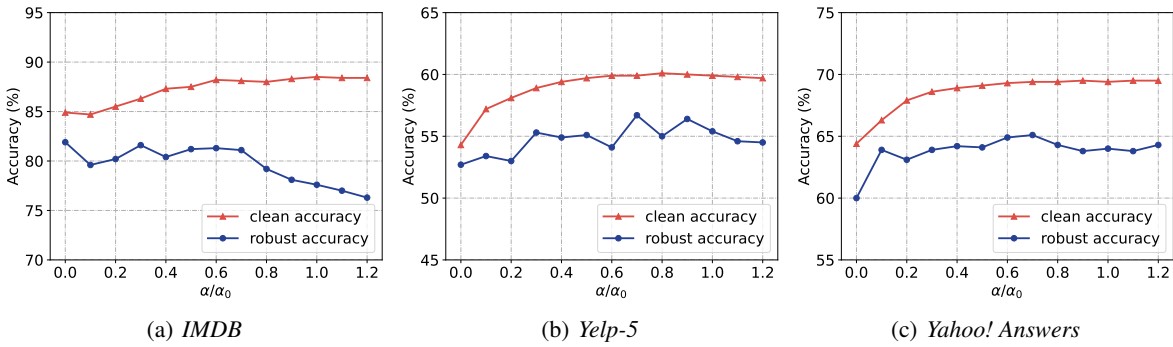

(a) *IMDB*    (b) *Yelp-5*    (c) *Yahoo! Answers*

Figure 2: The impact of hyper-parameter $\alpha$ on the performance of FTML on CNN models across the three datasets.

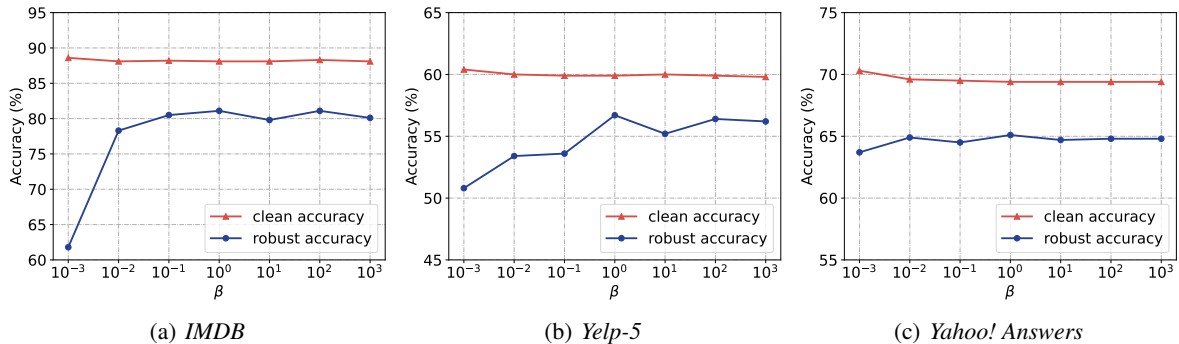

(a) *IMDB*    (b) *Yelp-5*    (c) *Yahoo! Answers*

Figure 3: The impact of hyper-parameter $\beta$ on the performance of FTML on CNN models across the three datasets.

or Chebyshev distance ($p = \infty$). The results are presented in Table 6. FTML with Manhattan distance achieves competitive robustness to FTML with Euclidean distance on three models. FTML with Chebyshev distance is inferior to the models trained with the other two distance metrics, but still enhances the robustness compared to the standard trained models (see results in Table 2). The reason is that Manhattan distance and Euclidean distance consider the difference of word vectors in all dimensions, while FTML with Chebyshev distance only updates the dimension with the largest difference in each training iteration, making the models hard to converge.

**Hyper-parameter Study.** FTML involves two hyper-parameters. $\alpha$ is used for constraining the distance between the anchor word and its non-synonyms in the word-level triplet loss. $\beta$ controls the weight of the word-level triplet loss in the overall training objective function.

In Figure 2, the *clean accuracy* denotes the classification accuracy (%) on the entire original testing set, while the *robust accuracy* denotes the classification accuracy (%) under the PWWS attack. With $\beta$ being fixed to 1, we vary $\alpha/\alpha_0$ from 0.0 to 1.2 to investigate how $\alpha$ influence the performance of FTML. When $\alpha = 0$, FTML only forces the words to be close to their synonyms without considering the non-synonyms, and the clean accuracy is the lowest.

Intuitively, the reason is that the existence of polysemous words will cause all the words that have at least one same meaning to be compressed together, making the model unable to distinguish them. With the increment of $\alpha$, the clean accuracy of the models rises considerably, while the robust accuracy fluctuates but first increases and then declines in general. Hence, we choose $\alpha = 0.7\alpha_0$ for the three datasets to achieve a proper trade-off on the clean accuracy and the robust accuracy.

Similarly, in Figure 3, we vary parameter $\beta = 10^i$ ($-3 \le i \le 3$) to explore its sensitivity with the other parameter $\alpha$ being fixed to $0.7\alpha_0$. We can observe that the clean accuracy tends to decay slightly with the increment of $\beta$. Also, FTML performs stably under the PWWS attack for a wide range of $\beta$ in $[10^0, 10^3]$ over all the three datasets. Therefore, we choose $\beta = 1$ to have a proper trade-off on the clean accuracy and the robust accuracy.

**Analysis on Supplementing BERT's Vocabulary.** For the recent pre-trained NLP models based on sub-words such as BERT, some actual words that do not exist in the model's vocabulary would be divided into several sub-words, making it hard to calculate the distance between the words in the embedding space for FTML. To solve this problem, we add some actual words into the vocabulary during the fine-tuning phase without affecting the pre-training phase.

Table 7: The influence of various number of actual words included in the vocabulary on BERT fine-tuing. *Robust accuracy* denotes the accuracy of FTML trained BERT models against PWWS attack. *Clean accuracy* denotes the accuracy of normally trained BERT models on the original test set.

| # actual words | 50,000 | 45,000 | 40,000 | 35,000 | 30,000 |
|---|---|---|---|---|---|
| Vocabulary size | 59,734 | 55,523 | 51,402 | 47,387 | 43,549 |
| Robust accuracy (%) | 81.2 | 77.3 | 75.7 | 68.9 | 64.7 |
| Clean accuracy (%) | 92.2 | 92.2 | 92.4 | 92.5 | 92.4 |

The common words are limited. Besides, many actual words have been included in the original vocabulary as sub-words. Thus, the vocabulary size would not be increased much. In our experiments, we only supplement the BERT's vocabulary to contain the top 50,000 words with the highest frequency in the dataset, which increases the vocabulary size from 30,522 to 59,734. To investigate the influence of the number of actual words on the performance of FTML, we fine-tune BERT by FTML with various numbers of actual words included in the vocabulary. We report the robust accuracy of FTML trained models against PWWS attack on the IMDB dataset. The results are summarized in Table 7. When we supplement the vocabulary to contain 30,000 actual words, the vocabulary size is 1.5 times larger than the original one, and the robustness has been significantly improved compared with normally fine-tuned BERT (16.6%). When the vocabulary size increases gradually, the robustness of the FTML trained model has been greatly improved.

To investigate the influence of the number of added words on the generalization, we adopt the expanded vocabulary to fine-tune the BERT model without FTML on the IMDB dataset. As shown in Table 7, the classification performance of BERT models fine-tuned with the expanded vocabulary varies from 92.2% to 92.5%, which is very close to the one with the original vocabulary (92.4%). It indicates that adding some additional words to the vocabulary during the fine-tuning stage does not affect the generalization of the BERT model. Interestingly, when we supplement the vocabulary to contain 35,000 actual words, the model even performs slightly better than the one with the original vocabulary.

## 5 CONCLUSION

In this work, we introduce a novel method termed *Fast Triplet Metric Learning (FTML)* to train models with robust word embeddings. Specifically, we incorporate the standard training with a word-level triplet loss, which pulls the words to be closer to their synonyms and pushes away their non-synonyms in the embedding space. Extensive experiments demonstrate that FTML achieves much higher robustness against various attacks than existing state-of-the-art baselines. FTML is efficient, making it easy to extend to large-

scale datasets and complex models. It is noted that the idea of how to learn a robust word embedding in FTML is general to any language. Once we have prepared a synonym dictionary for the given language, we can directly apply FTML to pull the words closer to their synonyms and push away their non-synonyms in the embedding space to train an adversarially robust model.

Our work exhibits great potential of improving the textual robustness through robust word embedding, which challenges the mainstream view of enhancing the robustness of the overall model against adversarial attacks and highlights the difference of model robustness on texts and images. We hope our work could inspire more works in this direction by considering the speciality of natural languages.

## Acknowledgements

This work is supported by National Natural Science Foundation of China (62076105) and International Cooperation Foundation of Hubei Province, China (2021EHB011).

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
