# OpenReview forum: "Robust Textual Embedding against Word-level Adversarial Attacks"
_auai.org/UAI/2022/Conference — UAI 2022 Poster_

### Official Review · Reviewer_h1s1 · 2022-04-12

**Q2(1) Originality/Novelty:** 2
**Q2(2) Significance/Impact:** 2
**Q2(3) Correctness/Technical Quality:** 3
**Q2(6) Clarity Of Writing:** 4
**Q6 Overall Score:** 5
**Q8 Confidence In Your Score:** 3

**Q1 Summary And Contributions:**

The paper proposes Fast Triplet Metric Learning (FTML) to improve robustness of NLP classification models. This adds a triplet loss to induce embeddings of synonyms to be close to each other and embeddings of random words to be far from each other. The authors state that this improves the robustness vs accuracy trade-off compared to baselines and approaches proposed in past work.

**Q2 Assessment Of The Paper:**

More detailed information regarding each of these aspects is given below:

**Q2(4) Quality Of Experiments (Optional):**

2: Fair: The experimental evaluation is weak: important baselines are missing, or the results do not adequately support the main claims.

**Q2(5) Reproducibility:**

1: Poor: Key details (e.g., proof sketches, experimental setup) are incomplete/unclear, or key resources (e.g., proofs, code, data) are unavailable.

**Q3 Main Strengths:**

* The paper is well written - the proposed improvements are well situated with related work, the proposed Fast Triplet Metric Learning is defined clearly.
* The experiments are adequate and there are enough ablations to justify the choices made e.g. choice of distance metric or choice of loss formulation (contrastive loss vs triplet loss), impact of hyper-parameter tuning


**Q4 Main Weakness:**

* The biggest weakness is the fact that at least since 2018-2019, NLP models rely on subwords based on BPE, WordPiece, SentencePiece tokenization, rather than actual words in a language.
* The proposed approach relies on actual words being used in the model's vocabulary which is sub-optimal and has not been used for the last few years of NLP research. e.g. The paper adds additional vocabulary terms to pre-trained BERT to adopt BERT to their approach. This is not a sustainable solution since it could lead to the vocabulary size blowing up and poor generalization.
* Second, it is unclear where the synonyms list comes from and how domain specific is it? e.g. would a generic synonyms list help improve robustness on an NLP task in the medical or legal domain?
* How extensible is the process of gathering synonym lists to other languages or domains?
* The paper is missing the potentially strong baseline of adding the adversarially perturbed data as augmented training data.

**Q5 Detailed Comments To The Authors:**

-

**Q7 Justification For Your Score:**

The paper is well-written and well-explained but I would give more weight to the important weaknesses highlighted above.

**Q9 Complying With Reviewing Instructions:**

1: Yes.

---

### Official Review · Reviewer_izw7 · 2022-04-12

**Q2(1) Originality/Novelty:** 2
**Q2(2) Significance/Impact:** 2
**Q2(3) Correctness/Technical Quality:** 3
**Q2(6) Clarity Of Writing:** 4
**Q6 Overall Score:** 5
**Q8 Confidence In Your Score:** 3

**Q1 Summary And Contributions:**

The authors present a method for training models with strong word embeddings called Fast Triplet Metric Learning (FTML) in this paper. They combine normal training with a word-level triplet loss, which pulls words closer to their synonyms while pushing nonsynonymous further away in the embedding space. Extensive testing shows that FTML achieves substantially stronger robustness against multiple attacks than existing SOTA baselines.

**Q2 Assessment Of The Paper:**

More detailed information regarding each of these aspects is given below:

**Q2(4) Quality Of Experiments (Optional):**

3: Good: The experimental evaluation is adequate, and the results convincingly support the main claims.

**Q2(5) Reproducibility:**

2: Fair: Key resources (e.g., proofs, code, data) are unavailable but key details (e.g., proof sketches, experimental setup) are sufficiently well-described for an expert to confidently reproduce the main results.

**Q3 Main Strengths:**

1. The paper is well-organized and clearly written.
2. The proposed method is simple and straightforward, and thus easy to implement. The additionally introduced overhead is little.
3. The performance of the method is excellent. The improvement over existing SOTA baselines is significant.

**Q4 Main Weakness:**

1. The motivation is not strong. This claim `We attribute the vulnerability of natural language processing models to the fact that similar inputs are converted to dissimilar representations in the embedding space, leading to inconsistent outputs.' needs more explanation or justification.

2. The technique contribution is limited. As the authors stated, triplet metric learning is commonly used for machine learning algorithms. In this paper, the technique contribution can be summarized as adding a triplet metric learning regularization term to the original loss. Besides, this method requires a pre-trained third-party embedding space.

**Q5 Detailed Comments To The Authors:**

The proposed method is simple and straightforward but has excellent performance. The technique contribution is limited.

**Q7 Justification For Your Score:**

The performance is good yet the technique contribution is limited. Thus I arrive at a borderline accept.

**Q9 Complying With Reviewing Instructions:**

1: Yes.

---

### Official Review · Reviewer_2Awt · 2022-04-12

**Q2(1) Originality/Novelty:** 3
**Q2(2) Significance/Impact:** 3
**Q2(3) Correctness/Technical Quality:** 3
**Q2(6) Clarity Of Writing:** 3
**Q6 Overall Score:** 7
**Q8 Confidence In Your Score:** 3

**Q1 Summary And Contributions:**

This paper proposes a novel robust training model for textual embedding, fast triple metric learning (FTML), to distinguish its representation from other samples for better robustness. The idea is to pull the words closer to their positive samples (synonyms) and push away their negative samples (non-synonyms) in the embedding space. Experiments show that FTML could significantly promote the model robustness against various advanced adversarial attacks with stable accuracies.

**Q2 Assessment Of The Paper:**

More detailed information regarding each of these aspects is given below:

**Q2(4) Quality Of Experiments (Optional):**

3: Good: The experimental evaluation is adequate, and the results convincingly support the main claims.

**Q2(5) Reproducibility:**

3: Good: Key resources (e.g., proofs, code, data) are available and key details (e.g., proofs, experimental setup) are sufficiently well-described for competent researchers to confidently reproduce the main results.

**Q3 Main Strengths:**

1. the general idea of FTML is important to learn robust textual embedding against word-level adversarial attacks;
2. experiments on three datasets show that FTML can be efficient and achieve better results.

**Q4 Main Weakness:**

1. not clear if the approach is applicable to other pretrained methods such as GPT-x style.
2. not clear yet how the approach can be generalized to other languages and what should be prepared beforehand and even how to do things of the other direction of from robust word embedding back to synonym finding.

**Q5 Detailed Comments To The Authors:**

1. leveraging synonyms and non-synonyms information for robust word embedding learning is interesting, how about the other direction of from robust word embedding learning to further back boost better classifications of synonyms and non-synonyms?
2. how to generalize your method to other languages? what should be prepared by hand or automatically?
3. besides bert/cnn/lstm, how shall your method be applied to gpt-x style generation methods and further downstream generation tasks?

**Q7 Justification For Your Score:**

1. the general idea of robust word embedding learning and its application to cnn/lstm/bert learning is solid and strong as testified by a list of datasets;
2. the motivation, method description, and experiments are well written and should bring impacts to the related research field.

**Q9 Complying With Reviewing Instructions:**

1: Yes.

---

### Decision · Program_Chairs · 2022-05-15

**Decision:**

Accept (Poster)

**Comment:**

Meta Review: All reviewers show positive opinions of this paper. AC read the paper (abstract+introduction), reviews, and authors' responses. After the response period, all reviewers kept their original ratings. AC recommends acceptance